

**Effects of Y-type spillway lateral contraction ratios on debris flow patterns**
**and scour features behind a check dam**

3               **Huayong Chen [1,2], Jinfeng Liu [1,2], and Wanyu Zhao [1,2]**

[1]Key Laboratory of Mountain Hazards and Earth Surface Process Chinese Academy of Sciences (CAS) , Chengdu 610041, China
[2]Institute of Mountain Hazards and Environment, CAS, Chengdu 610041, China
*Correspondence to*: Huayong Chen (hychen@imde.ac.cn)
**Abstract.** Debris flows often cause devastating damage to property and can injure or kill residents in
mountainous areas. The construction of check dams in debris flow valleys is considered a useful strategy for
mitigating the damages downstream. In this paper, a new type of spillway structure with lateral contraction was
proposed to distribute debris flows after the check dam storage filled up. Four different lateral contraction ratios
of the spillway were considered in experiments that investigated debris flow patterns, scour characteristics, and
energy dissipation rates when debris flows passed through the spillway. The results indicated that lateral
contraction considerably influenced the extension of debris flow nappes. The drop length of the nappe at $\eta$=0.7
($\eta$ means lateral contraction ratio) was approximately 1.4 times larger than at $\eta$=0.4. The collision, friction, and
mixing forces between the debris flow nappes and debris flows in downstream plunge pools dissipated much of
the debris flow kinetic energy, ranging from 42.03% to 78.08% at different contraction ratios. Additionally,
based on a dimensionless analysis, an empirical model was proposed to predict the maximum scour depth behind
the check dam. It indicated that the results calculated by the model exhibited good agreement with the
experimental results.
**1    Introduction**
Debris flows are formed by poorly sorted, water-saturated materials that mobilize in upstream regions of
valleys and surge down slopes in response to gravitational attraction (Iverson,1997). Large scale debris flows
were triggered by intensive rainfalls after the "5.12" Wenchuan Earthquake, including the Zhouqu debris flow,





the Wenjia gully debris flow, and the Hongchun gully debris flow (Wang, 2013; Yu et al. , 2013; Tang et al.,
2015). On August 8, 2010, a large debris flow occurred in the Luojiayu gully, northern Zhouqu County, Gansu
Province. The flow destroyed six villages, blocked the Bailongjiang River, resulting in the formation of a lake
that inundated over half of Zhouqu County, and displaced or killed 1765 people (Cui et al., 2013). Usually,
large-scale debris flow events involve substantial erosion upstream (Ni et al., 2012; Yu et al., 2013), and large
volumes of solid materials are transported from the formation region to downstream areas by debris flows.
The construction of check dams is considered one of the most effective ways to store solid materials and
control soil erosion in a valley. This structural counter-measure is commonly used to stabilize bank slopes,
flatten the gradients of valleys, reduce flow velocity, and decrease the peak-discharge of debris flows (Lenzi,
2002; Mizuyama, 2008; Remaître et al., 2008; Remaitre and Malet, 2010). Two main types of check dams are
applied to control debris flows (i.e., closed-type and opened-type). Opened-type dams trap boulders, cobble, and
gravel, allowing small particles, fine sediments, and water to pass through the dams (Abedini et al., 2012).
Closed-type damns not only trap the coarse particles but also retain most small particle materials (Heumader,
2000; Lien, 2003). Generally, the dam storage volume of a closed-type check dam is quickly filled with debris
flow material when a large debris flow occurs. The sequent debris flows directly overflow the check dam, which
can lead to serious scour on and around the foundation of the check dam (Figure 1).
Flow patterns and scour caused by the discharge of clear water or sediment flows has been well studied in
hydraulic engineering. The characteristics of free-falling nappes behind the spillway of a gravity dam were
investigated and the drop length of the free jet was predicted based on the energy equation in which the energy
dissipation was neglected at two chosen cross-section (Toombes et al., 2008). Experimental investigations of
aeration associated with overflow dams with curved surfaces were carried out, and empirical correlations
predicting the aeration efficiencies of these differently shaped spillways were developed (Chu et al., 2014). An



interpolation formula for predicting scour depth was proposed based on experimental data. It indicated that the
maximum scour increased with increasing discharge and decreased with increasing downstream tail water depth
(Adduce et al., 2005). In addition to the discharge and downstream tail water depth, the characteristic grain size
and the plunge angle were also considered for scour depth prediction (Bormann and Julien, 1991). Considerable
attention has been given to the flow patterns and scour caused by clear flows or sediment flows behind dams.
However, few studies have investigated the debris flow patterns and scour features behind check dams (Pan et al.,
2013), especially for spillway structures with lateral contraction. The flow patterns and scour features caused by
debris flows are different from those caused by clear water or sediment flows due to different flow densities,
cohesion, and particle volume concentrations. The investigation on characteristics of debris flows discharging
and scouring with Y-type spillway can help us better understand the interaction between debris flows and the
erodible solid materials, which can also help us to find out better methods for debris flow mitigation in some
serious geology conditions.

In this paper, a new spillway structure with lateral contraction was proposed. Experiments with different

spillway contraction ratios were conducted to study the characteristics of debris flow nappes and scour after
debris flows overflowed the check dam. For each experimental test, video cameras were used to record the
trajectories of debris flow nappes. The energy dissipation rate was analyzed due to the varying lateral contraction
ratios. Finally, an empirical model based on dimensionless analysis was proposed to predict the maximum scour
depth behind the check dam.
**2    Experimental setup**

The experiments were performed at the Dongchuan Debris Flow Observation and Research Station (DDFORS)

in Dongchuan District, Yunnan Province, China. Generally, the experimental flume consisted of a hopper, a gate,
a rectangular channel, and the downstream erodible bed (Figure 2a). The rectangular channel was approximately





4.0 m long, 0.4 m wide, and 0.4 m high, with a slope of 8° (Figure 2b). A check dam made of steel material was
located at the end of the rectangular channel. The shape of the spillway inlet was a 0.20 m wide by 0.10 m high
rectangle. The outlet was shaped like a capital letter 'Y'. The top width of the outlet was equal to that of the inlet.
The bottom width ranged from 0.06 m to 0.12 m due to the different contraction ratios of the spillway. The
dimensions of the spillway are shown in Figure 2c.
The lateral contraction ratio $\eta$ is defined as follows:

$$\eta = \frac{B-b}{B} \qquad (1)$$

where $B$ is the width of the spillway inlet and $b$ is the width of the spillway outlet. When $b=B$, $\eta=0$.
The storage of the check dam was filled with the solid materials from Jiangjia ravine, with a slope of 3°. The
diameter of the solid material was smaller than 20.0 mm. Its particle size distribution is shown in Fig. 3. Particle
size distribution may affect the debris flow density and flow motion along the channel. The solid materials used
in this experiment was prepared according to the sample of typical debris flows and excluded particles larger
than 20.0 mm due to the limitations of the experimental conditions. The diameter of the solid materials in the
erodible bed was also smaller than 20.0 mm. In addition, the clay and fine particles (smaller than 1.0 mm) were
excluded to avoid the effects of matric suction on the development of the scour hole. The particle size
distribution of the erodible bed materials is also shown in Figure 3.
In each experimental test, a laser range finder (LRF) was set at the end of the erodible bed to monitor the
depth of the debris flow during the entire process, as shown in Figure 4. The LRF measured the distance
between the original bottom and the laser receiver. When debris flows flowed over the channel bottom, the
LRF measured the distance between the debris flow surface and the laser receiver. The distance difference
was the flow depth. The measurement range of the LRF was up to 30.0 m, with an accuracy of ±0.001 m. The
elevation difference between the initial position and the flow surface was the measured flow depth. An example





of the measured results is shown in Figure 5. It reveals that although the debris flow process is not steady over
time, the debris flow over a short period can be considered steady flow. Therefore, the energy conservation
equation derived based on the steady flow assumption can be applied to analyze the energy dissipation rate of a
debris flow.
**3    Experimental results and analysis**
**3.1    Flow patterns of different contraction ratios**
When debris flows overflowed the spillway with a high lateral contraction ratio ($\eta$=0.7), the flow depth and
velocity increased dramatically. The debris flow nappe clearly extended in the flow direction. Furthermore, the
debris flows near both side wall, which were forced to change direction by the walls, collided at the outlet when
the debris flows overflowed from the spillway (Figure 6a). Decreasing the lateral contraction ratio caused the
flow depth and velocity to decrease at the same flow discharge. Therefore, the drop length of the debris flow
nappe decreased in the flow direction. The drop length at $\eta$=0.7 was approximately 1.4 times than at $\eta$=0.4
(Table 1). Lateral contraction not only affected the drop length but also broadened the nappe width due to the
collision of debris flows at the outlet (Figure 6b-d). When $\eta$=0.5, the broadening ratio $\kappa$ ($\kappa$ is the ratio of nappe
width to the outlet width) reached its maximum value ($\kappa$=2.93 in Table 1). The nappe width was equal to that of
the spillway ($\kappa$=1.0) when there was no lateral contraction at the spillway.
If debris flows flowing out of the spillway are considered free-motion point masses under the influence of
gravity, the trajectory of a debris flow nappe can be expressed as follows (Figure 7):

$$y = xtg\varphi + \frac{g}{2v_1^2 \cos\varphi^2} x^2 \qquad (2)$$

$$x = \frac{v^2}{g} \cos\varphi \left( \sqrt{\sin^2\varphi + \frac{2gy}{v^2}} - \sin\varphi \right) \varphi \geq 0 \qquad (3)$$

When  $\varphi = 0$ , equation (2) simplifies to equation (3):


$$x = \sqrt{\frac{2v^2 y}{g}} \qquad (4)$$

where $v$ is the initial velocity of the debris flow flowing out of the spillway, $\varphi$ represents the angle of the
initial velocity in the horizontal direction, and $y$ is the water elevation difference.
Equation (3) indicates that the nappe extension in the horizontal direction '$x$' is proportional to the initial
velocity $v$ and square root of the water elevation difference $y$. From Fig. 6 and Table 1, we found that when
$\eta$=0.7, the nappe extension was longest in the flow direction. From this point of view, a high lateral contraction
ratio increased the distance between the plunge point and the dam toe, which effectively protected the dam
foundation from scouring. The hydraulic characteristics of the nappe away from the spillway at different lateral
contraction ratios were shown in Figure 8 and Figure 9. Figure 8 indicates that increasing the lateral contraction
ratio decreased the width of the debris flow nappe. Furthermore, the higher lateral contraction of the spillway
strengthened the collision between flows at the spillway outlet. Air bubbles were entrained in the debris flows
when the continuum of the debris flows was broken. Figure 9 shows the extent of the debris flow nappes. The
distribution of the flow velocity in the vertical direction at the outlet increased with increasing flow depth due to
the effects of boundary friction. Therefore, the longest flow nappes were formed by the debris flows with
relatively large velocities at the flow surface.
**3.2**     **Debris flow scour features behind the check dam**
The scour features of debris flows behind the check dam represent one of the most important indexes, which
determines the scour depth at the dam foundation. Figure 10 shows the effects of lateral contraction on the
formation of scour holes in an erodible bed. For the same curvature of the spillway surface, decreasing the
contraction ratio decreased the maximum scour depth and caused the location of the maximum scour point to
shift toward the dam toe due to the decreased debris flow velocity. The maximum scour depth and its location





farther from the dam toe for $\eta=0.7$ were approximately 1.3 and 1.4 times, respectively, larger than for $\eta=0.4$.
Although a high lateral contraction ratio extended the debris flow nappe, it also increased the scour depth in the
erodible bed to some extent.
**3.3    Energy dissipation at different contraction ratios**
Generally, different energy dissipaters such as the plunge pool (Pagliara et al., 2010; Duarte et al., 2015) or
step-pool systems (Yu, 2007; Wang et al., 2009; Wang et al., 2012) are required to dissipate the kinetic energy of
the surplus flow and prevent the dam foundation and riverbed from scouring when sudden changes to the
channel slope occur. The energy dissipation process of the check dam was estimated using the Bernoulli equation
(4). The rationale behind using this equation was previously mentioned.
The Bernoulli equation between two reference cross-sections is written as follows:

$$Z_1 + h_1 + \alpha_1 \frac{v_1^2}{2g} = Z_2 + h_2 + \alpha_2 \frac{v_2^2}{2g} + h_w \tag{5}$$

If $\Delta Z = Z_1 - Z_2$, then equation (4) can be transformed into equation (5):

$$\Delta Z + h_1 + \alpha_1 \frac{v_1^2}{2g} = h_2 + \alpha_2 \frac{v_2^2}{2g} + h_w \tag{6}$$

The energy dissipation coefficient $\zeta$ can be expressed as follows:

$$\zeta = 1 - \frac{h_2 + \dfrac{v_2^2}{2g}}{\Delta z + h_1 + \dfrac{v_1^2}{2g}} \tag{7}$$

where $Z_1$ and $Z_2$ are the elevations of reference cross-sections #1 and #2 (Figure. 2b), respectively; $h_1$ and $h_2$
are the depths of debris flows at reference cross-sections #1 and #2, respectively; $v_1$ and $v_2$ are the velocities of
the debris flows at references cross-sections #1 and #2, respectively; $\alpha_1$ and $\alpha_2$ are the kinetic energy correction
coefficients ($\alpha_1=\alpha_2=1$) (Adamkowski et al., 2006); $\Delta Z$ is the elevation difference between the two reference



cross-sections; and $h_w$ is the water head loss.
Table 2 indicates that the collision and friction forces between the debris flow nappes and debris flows in the
plunge pool dissipated the kinetic energy of the flows, ranging from 42.03% to 78.08% at different contraction
ratios. In the case of $V$=0.16 m³, the energy dissipation rate decreased gradually when the contraction ratio
changed from 0.7 to 0.4 because the high contraction ratio increased the number of debris flow collisions when it
passed through the spillway. In the cases of $V$=0.10 m³ and $V$=0.06 m³, the energy dissipation rate also decreased
with decreasing the contraction ratios except at $\eta$=0.4. The mean value of the energy dissipation rate
demonstrated a good positive correlation between the energy dissipation rate and the lateral contraction ratio. In
addition, for the same contraction ratio, the energy dissipation rate increased gradually with decreasing debris
flow scale.
**3.4      The empirical equation for estimating the maximum scour depth**
Many empirical equations have been proposed to predict the maximum scour depth over the last several
decades (Bormann and Julien, 1991; Zhou, 1991; Adduce et al., 2005; Pan et al., 2013). The main parameters
include the unit discharge, characteristic particle size of the erodible bed, water elevation difference and clear
water and debris flow densities. However, most of the empirical equations (Li and Liu, 2010) neglect
dimensional homogeneity (the empirical equations should be dimensionally homogeneous). For new type of
spillway, the lateral contraction ratio is an important parameter for predicting the maximum scour depth. For a
debris flow, the maximum scour depth is mainly determined by the following parameters:

$$h_d = f(q, g, \rho_d, \rho_w, d_{90}, \eta......) \tag{8}$$

where $h_d$ is the maximum scour depth, $q$ is the unit discharge of the debris flow, $g$ is the acceleration due to
gravity, $\rho_d$ and $\rho_w$ are the debris flow density and clear water density, respectively (two debris flow densities
were considered, including, $\rho_d$=1200kg/m³ and $\rho_d$=1500kg/m³), $d_{90}$ is the characteristic particle size for erodible





bed materials , and $\eta$ is the lateral contraction ratio.
Based on a dimensional analysis, the dimensionless parameters with clear physical meanings are developed as
follows:

$$\frac{h_s}{d_{90}} = k \left( \frac{q}{d_{90}\sqrt{gd_{90}}} \right)^{a1} \left( \frac{\rho_d}{\rho_w} \right)^{a2} (1-\eta)^{a3} \tag{9}$$

where $h_s/d_{90}$ is dimensionless scour depth, $k$ is a coefficient, ai is an index (i=1, 2, 3), $\dfrac{q}{d_{90}\sqrt{gd_{90}}}$ is the
dimensionless discharge, and $\rho_d/\rho_w$ is the dimensionless density.
According to the experimental data, the regression equation can be expressed as follows:

$$\frac{h_s}{d_{90}} = 3.15 \left( \frac{q}{d_{90}\sqrt{gd_{90}}} \right)^{0.51} \left( \frac{\rho_d}{\rho_w} \right)^{-0.1363} (1-\eta)^{0.7583} \tag{10}$$

The regression equation suggests that the flow density had relatively small effects on the depth of the scour
hole. However, the debris flow discharge and the lateral contraction had strong effects on the maximum depth of
the scour hole, which directly determined the kinetic energy of the flow in the downstream erodible bed. The
validation tests were also performed using the physical experimental model shown in Figure 2, but under
different conditions. Additional experimental data provided in the literature (Ben and Mossa, 2006) were used to
verify the reliability of the regression equation. The predicted results exhibited good agreement with the
experimental results. The absolute error was smaller than 15.0% in most cases, as shown in Figure 11.
**4    Conclusions**
The characteristics of debris flows overflowing the new type of spillway were analyzed at different lateral
contraction ratios. The energy dissipation rate and an empirical model for predicting the maximum scour depth
were also studied in this paper. The following conclusions were drawn from this analysis:
1)    Flow patterns were mainly determined by the lateral contraction ratio. At a high lateral contraction ratio, the

spillway effectively extended the debris flow nappe and increased the distance between the plunge point

and the dam toe. The drop length of the nappe at $\eta$=0.7 was approximately 1.4 times higher than that at



$\eta$=0.4.
2)    The plunge pool behind the check dam inevitably dissipated the kinetic energy of the debris flow after

overflowing the check dam. The collision and friction between the debris flow nappe and the debris flow in

the plunge pool dissipated the kinetic energy of the flow, ranging from 42.03% to 78.08% at different

contraction ratios. Generally, increasing the contraction ratio increased the energy dissipation rate at the

same debris flow scale.

3)    An empirical model was proposed to predict the maximum scour depth behind the check dam. The results

indicated that the predicted results exhibited good agreement with the experimental results. The absolute

error was smaller than 15.0% in most cases.

**Acknowledgments**
The study results presented in this paper were supported by the National Natural Science Foundation of China
(Grant No.51209195), the Key Research Program of the Chinese Academy of Sciences (Grant No.
KZZD-EW-05-01), the Science Technology Service Network Initiative, Chinese Academy of Sciences (Grant No.
KFJ-EW-STS-094), and the Key Laboratory of Mountain Hazards and Earth Surface Process, Chinese Academy
of Sciences.
**List of symbols**

ai  =  The index for the dimensionless parameter (-)
$b$  =  The width of the spillway outlet(m)
$B$  =  The width of the spillway inlet (m)
$d_{90}$  =  The characteristic particle size for erodible bed materials (m)
$k$  =  The coefficient for the dimensionless equation (-)
$h_1$  =  The depth of debris flows at reference cross-sections #1 (m)
$h_2$  =  The depth of debris flows at reference cross-sections #2 (m)
$h_d$  =  The maximum scour depth (m)
$h_w$  =  The water head loss (m)
$g$  =  The acceleration of gravity (m/s$^2$)
$q$  =  The unit discharge of the debris flow (m$^3$/s)
$v$  =  The initial velocity of the debris flow flowing out of the spillway(m/s)
$v_1$  =  The velocity of debris flows at reference cross-sections #1 (m/s)
$v_2$  =  The velocity of debris flows at reference cross-sections #2 (m/s)
$V$  =  The scale of debris flow in the experiments (m$^3$)





$x$ = Trajectory in the horizontal direction (m)
$y$ = The water elevation difference (m)
$Z_1$ = The elevation of reference cross-sections at #1 (m)
$Z_2$ = The elevation of reference cross-sections at #2 (m)
$\Delta z$ = The elevation difference between the two reference cross-sections (m)

Greek letters

$\alpha_1$ = The kinetic energy correction coefficient for $v_1$ (-)
$\alpha_2$ = The kinetic energy correction coefficient for $v_2$ (-)
$\rho_d$ = The density of debris flows (kg/ m$^3$)
$\rho_w$ = The density of clear water (kg/ m$^3$)
$\zeta$ = The energy dissipation coefficient(-)
$\eta$ = The lateral contraction ratio(-)
$\varphi$ = The angle of the initial velocity in the horizontal direction(°)





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





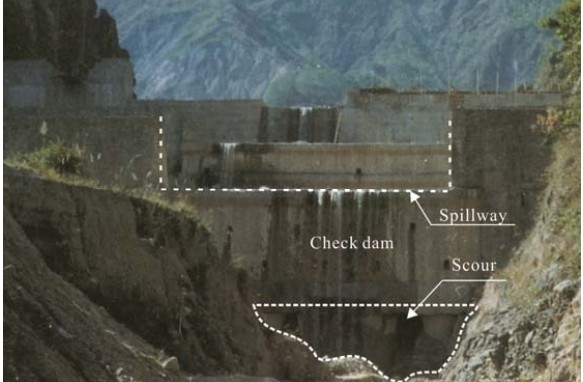


**Fig. 1.** An example of foundation scour behind a check dam





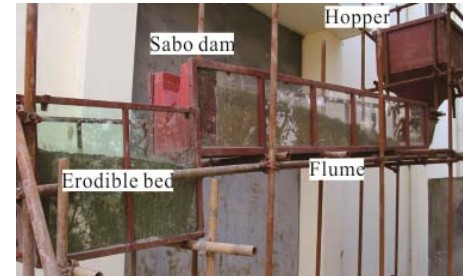

(a) Photograph of the experimental setup  (b) Schematic diagram of the experimental setup (unit: cm)

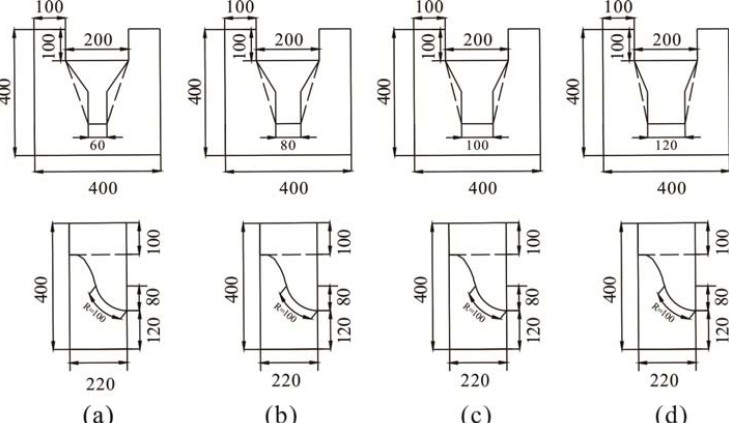


(c) The structure and dimensions of the spillway (unit: mm). Four different lateral contraction ratios were
considered in the experiments: (a) $B$=200.0 mm, $b$=60.0 mm, $\eta$=0.7; (b) $B$ =200.0 mm, $b$ = 80.0 mm, $\eta$=0.6; (c) $B$
=200.0 mm, $b$ =100.0 mm, $\eta$=0.5; (d) $B$ =200.0 mm, $b$ =120.0 mm, $\eta$=0.4. The bottom of the spillway was
formed by a compound curve surface (a simple curved segment and a circular segment: radius $R$=100.0 mm,
radius angle $\delta$=75°).
**Fig. 2.** Experimental setup






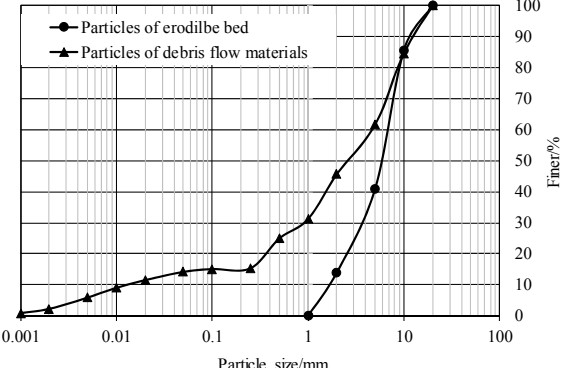


**Fig. 3.** The particle size distribution of samples for the debris flows and erodible bed




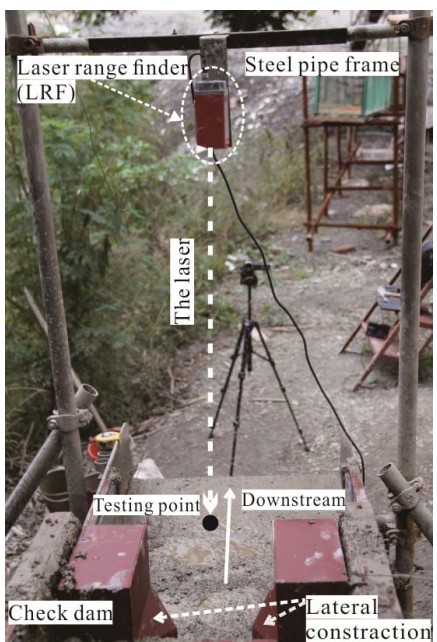


**Fig. 4.** Photograph of the LRF system (the photograph was taken in the downstream direction)




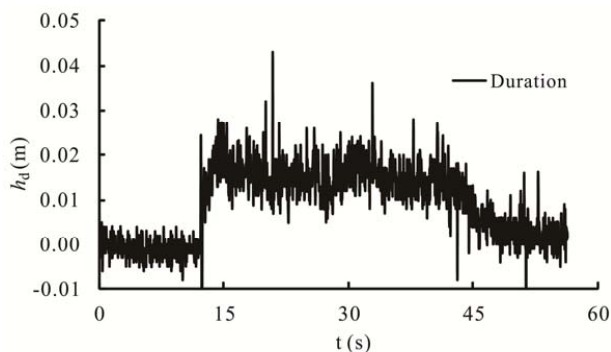


**Fig. 5.** An example of a debris flow duration monitored by the LRF




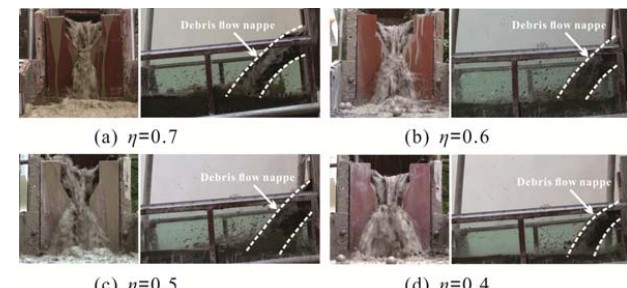


**Fig. 6.** Various debris flow patterns at different lateral contraction ratios (the pictures on the left were taken
from a downstream view)






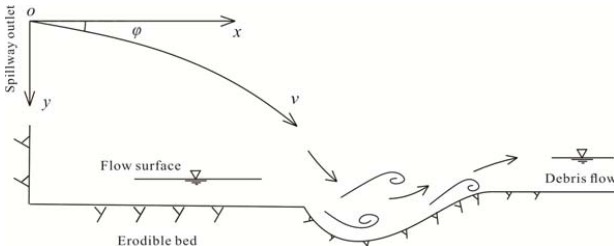


**Fig. 7.** A diagram of dynamic parameters of debris flows






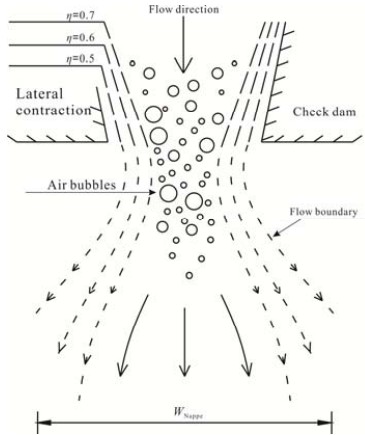


**Fig. 8.** The transverse expansion of a debris flow nappe at different lateral contraction ratios







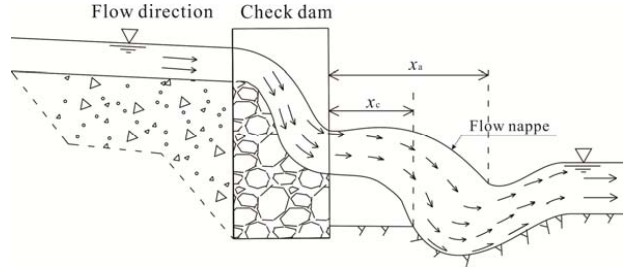


**Fig. 9.** The trajectory of a debris flow nappe




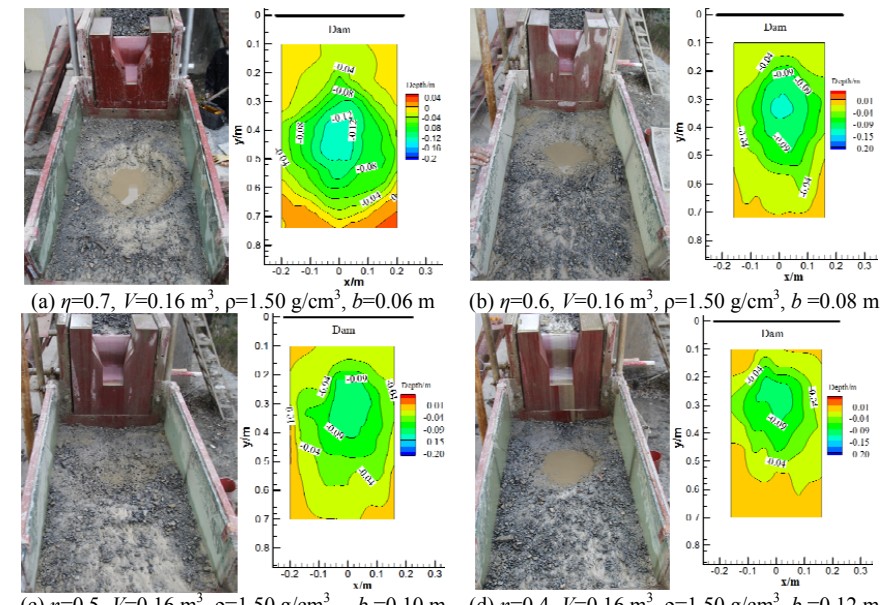

(a) $\eta$=0.7, $V$=0.16 m$^3$, $\rho$=1.50 g/cm$^3$, $b$=0.06 m    (b) $\eta$=0.6, $V$=0.16 m$^3$, $\rho$=1.50 g/cm$^3$, $b$=0.08 m

(c) $\eta$=0.5, $V$=0.16 m$^3$, $\rho$=1.50 g/cm$^3$,    $b$=0.10 m    (d) $\eta$=0.4, $V$=0.16 m$^3$, $\rho$=1.50 g/cm$^3$, $b$=0.12 m

**Fig. 10.** The shapes of the scour hole behind the check dam ($V$=0.16 m$^3$, $\rho$=1.50 g/cm$^3$)




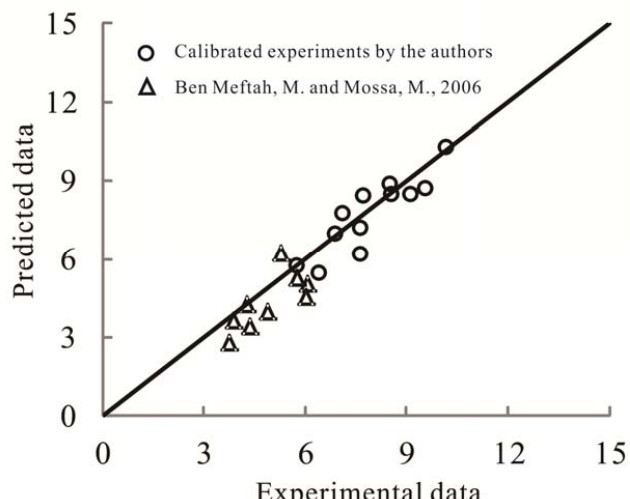


**Fig. 11.** Comparison between predicted data and experimental ones





**Table 1.** The main parameters of the debris flow nappe for different contraction ratios

| Items | (a) | (b) | (c) | (d) |
|---|---|---|---|---|
| Width of the outlet $b$/mm | 60.0 | 80.0 | 100.0 | 120.0 |
| Lateral contraction ratio $\eta$ | 0.7 | 0.6 | 0.5 | 0.4 |
| Width of the nappe $W_{\text{Nappe}}$ /mm | 137.2 | 231.6 | 292.6 | 320.6 |
| Broadening ratio $\kappa$($\kappa= W_{\text{Nappe}}$ /$b$) | 2.29 | 2.90 | 2.93 | 2.67 |
| Length of the nappe away from the outlet $x_a$/m | 0.43 | 0.34 | 0.33 | 0.31 |
| Length of the nappe close to the outlet $x_c$/m | 0.25 | 0.21 | 0.21 | 0.18 |

Notes: B is constant for each spillway type (B =200.0 mm)





**Table 2.** The energy dissipation rates at different contraction ratios

| Scales | Density ($\rho$=1.50 g/cm$^3$) | | | |
|---|---|---|---|---|
| | $\eta$=0.7 | $\eta$=0.6 | $\eta$=0.5 | $\eta$=0.4 |
| $V$=0.16 m$^3$ | 66.43% | 57.48% | 52.34% | 42.03% |
| $V$=0.10 m$^3$ | 75.37% | 72.94% | 60.58% | 67.97% |
| $V$=0.06 m$^3$ | 78.08% | 73.70% | 63.61% | 71.75% |
| Mean value | 73.29% | 68.04% | 58.84% | 60.58% |
