# Peer review of "Effects of Y-type spillway lateral contraction ratios on debris flow patterns and scour features behind a check dam"

_Natural Hazards and Earth System Sciences, 2016_

## Referee Comment (RC1) · Anonymous Referee #1 · 12 Jul 2016

The paper by Chen et al. address relevant scientific and technical questions and presents new data of new concepts. They are up to international standards. The scientific methods and assumptions are valid and outlined cleary. The results are sufficient to support the interpretations and conclusions. Some questions remain still unanswered and should be at least considered in the discussion. The abstract provides a concise, complete summary of the work done. Results are clearly presented. The mathematical formulae, symbols, abbreviations and units are correctly defined and used. Figures could be slightly improved. The authors give proper credit to previous amd related work. Own contributions are well indicated. Structure and length of the paper is adequate. Technical language and the English is of good quality and understandable.

some remarks concerning the text: p.1, line 2 not sure, if "downriver of a check dam" would better describe the exact location of the scour

p.1, line 11 in cases where debris flow is used in a word composition (e.g. debris-flow pattern, debris-flow nappe) I learned, that there is a hyphen between debris and flow Please check the manuscript accordingly

p.2, line 29 more common is "initiation zone", not "formation region"; delete "by debris flow" at the end of the sentence, it's an unnecessary repetition.

p.3, line 56/57 not really clear, what this sentence means. Do you mean that the proposed geometry of such spillways is something that should be used especcially for torrents with high sediment disposability?

p.3, line 58 "is" instead of "was"

p.8, line 165 are the values for the density of the debris-flow densities measured values or assumptions? Both values seems to me more valid for hyperconcentrated flows. I would espect values in the order of 1700 - 1900 kg/mˆ3

Figures:

**1: indicate flow direction and exchange the word "behind" with "downriver of"**

**2a: Sabo dam is never use in the text. Use check dam or replace check dam with sabo dam in the text**

**5: desribe it as "debris-flow hydrograph". If your LRF gave you min, max and mean values, you could perhaps exolain the outliers. And: this hydrograph does not really show a typicall steep front of a debris flow. It looks more like a hyperconcentraded flood. Again: add information on the sampling rate of the device**

**6: add an arrow to show flow directions. Very small images. Perhaps increase contrast.**

General remarks:

Scaling effects are not discussed. Please add a section to explain how the results of the experiments can be use in real dimensions. What is the Froude number of your experiments?

I miss a sensitivity study on different debris-flow mixtures (e.g. higher densities, water content variations)

I miss information on the LRF. What is the sampling rate (in Hz) of the device? How are splashing effects handled?

What would happen, if there is driftwood involved? Did you test that or what do you expect in such a case?

Can you say something about abrasion rates and the expected life time of such structures?

On the whole a very interesting and promising paper with nice results and an auspicious spillway design!

---

## Author Comment (AC1) · 10 Aug 2016

1. p.1, line 2 not sure, if "downriver of a check dam" would better describe the exact location of the scour. Answer: Thanks very much for the reviewer's com-ment. The phrase "behind a check dam" has been replaced by "downriver of a check dam" for better description of the exact scour loca-tion.

2. p.1, line 11 in cases where debris flow is used in a word composition (e.g. debris-flow pattern, debris-flow nappe) I learned, that there is a hyphen between debris and flow Please check the manuscript accordingly. Answer:A hyphen was added between debris and flow in a sentence throughout the manuscript where "debris flow" was used as attributive.

[Figure]

3. p.2, line 29 more common is "initiation zone", not "formation region"; delete "by debris flow" at the end of the sentence, it's an unnecessary repetition. Answer: The wrong phrases in the sentence have been revised according to the reviewer's comments.

4. p.3, line 56/57 not really clear, what this sen-tence means. Do you mean that the proposed geometry of such spillways is something that should be used especcially for torrents with high sediment disposability? Answer: To avoid misunderstanding, the sentence has been modified in the manuscript.

5. p.3, line 58 "is" instead of "was". Answer: The word "was" has been replaced by "is" in this sentence.

6. p.8, line 165 are the values for the density of the debris-flow densities measured values or assumptions? Both values seems to me more valid for hyperconcentrated flows. I would espect values in the order of 1700 - 1900 kg/mËĘ3. Answer: The flow densities were measured after de-bris-flow samples were taken. Frankly, as for debris flows the flow density in our experi-ments seems lower. The experimental analysis here is considered to be the preliminary achievements. The authors appreciate the re-viewer's valuable suggestions to carry out more experiments involving debris-flow den-sities in the order of 1700 - 1900 kg/mËĘ3 in the future.

7. #1: indicate flow direction and exchange the word "behind" with "downriver of". Answer: The word "behind" has been replaced by "downriver of".

8. #2a: Sabo dam is never use in the text. Use check dam or replace check dam with sabo dam in the text. Answer: The word "Sabo dam" has been replaced by "check dam".

9. #5: desribe it as "debris-flow hydrograph". If your LRF gave you min, max and mean values, you could perhaps explain the outliers. And: this hydrograph does not really show a typicall steep front of a debris flow. It looks more like a hyperconcentraded flood. Again: add infor-mation on the sampling rate of the device. Answer: The caption of Figure 5 was changed to "de-bris-flow hydrograph". The information on the sampling rate of the device was added in line 90, page 5.

10. #6: add an arrow to show flow directions. Very small images. Perhaps increase contrast. Answer: An arrow in each figure was added to show debris-flow directions in Fig.6.

11. Scaling effects are not discussed. Please add a section to explain how the results of the ex-periments can be use in real dimensions. What is the Froude number of your experiments? Answer: The scaling effects are discussed in the re-vised vision in lines 207-215, page 11. The Froude number in our experiment ranged from 1.14 to 1.16. It meant that the debris flows in the experiments were supercritical flow (in lines91-92, page 5).

12. I miss a sensitivity study on different de-bris-flow mixtures (e.g. higher densities, water content variations) Answer: The variation of debris-flow density (dif-ferent debris-flow mixtures) on scour depth was added in lines 135-141, page 7.

13. I miss information on the LRF. What is the sampling rate (in Hz) of the device? How are splashing effects handled? Answer: The information on the LRF was given in lines 89-90, page 5 and the sampling rate (Frequency) was added in line 90, page 5.

14. What would happen, if there is driftwood in-volved? Did you test that or what do you ex-pect in such a case? Answer: Debris flows with driftwood will speed up the blockage and jamming of a check dam. Provided that driftwood is involved in our experiments, the check dam will capture driftwood when it passed through the spill-way with debris flows. The subsequent de-bris flows will overflow from the check dam crest once the spillway is blocked by the driftwood, which will cause scour downriver of a check dam. The debris flows with driftwood was not considered in the current experiments, but definitely the reviewer has raised a very important question. The related experiments will be carried out to investigate the behaviour of debris flows with driftwood and its scour feature in the future.

15. Can you say something about abrasion rates and the expected life time of such structures? Answer: Abrasion occurs due to the interaction be-tween solid particles in debris flows and the boundary of hydraulic structures. For a spill-way with curved bottom, the reaction of cen-trifugal force exerting on spillway bottom enhance the interaction between the solid par-ticles and the bottom (a component of the reaction force has the same direction as the gravitational force of debris flows near the outlet of the spillway). Although abrasion phenomenon is common, it is difficult to quantify the abrasion rate during an episode of debris flows. Abrasion may be one of the factors lead to the damage of spillway with lateral contrac-tion. However, some methods can be taken to mitigate the abrasion damage of such struc-tures by using anti-abrasion materials, or add the protecting layer. The check dam with lat-eral contracted spillway, like other check dams, the expected life time mainly depends on the debris-flow scales, flow velocity, particle concentration, etc.

Please also note the supplement to this comment:
http://www.nat-hazards-earth-syst-sci-discuss.net/nhess-2016-189/nhess-2016-189-AC1-supplement.pdf

**Supplement:**

We feel very grateful to the reviewer who has given us the valuable suggestions and comments for our paper. We have revised our manuscript accordingly.

Huayong Chen

Responses to the reviewer' comments:

| Comments of Anonymous Referee #1: | Author's Reply |
|---|---|
| **1.** p.1, line 2 not sure, if "downriver of a check dam" would better describe the exact location of the scour. | Thanks very much for the reviewer's comment. The phrase "behind a check dam" has been replaced by "downriver of a check dam" for better description of the exact scour location. |
| **2.** p.1, line 11 in cases where debris flow is used in a word composition (e.g. debris-flow pattern, debris-flow nappe) I learned, that there is a hyphen between debris and flow Please check the manuscript accordingly. | A hyphen was added between debris and flow in a sentence throughout the manuscript where "debris flow" was used as attributive. |
| **3.** p.2, line 29 more common is "initiation zone", not "formation region"; delete "by debris flow" at the end of the sentence, it's an unnecessary repetition. | The wrong phrases in the sentence have been revised according to the reviewer's comments. |
| **4.** p.3, line 56/57 not really clear, what this sentence means. Do you mean that the proposed geometry of such spillways is something that should be used especcially for torrents with high sediment disposability? | To avoid misunderstanding, the sentence has been modified in the manuscript. |
| **5.** p.3, line 58 "is" instead of "was". | The word "was" has been replaced by "is" in this sentence. |
| **6.** p.8, line 165 are the values for the density of the debris-flow densities measured values or assumptions? Both values seems to me more valid for hyperconcentrated flows. I would | The flow densities were measured after debris-flow samples were taken. Frankly, as for debris flows the flow density in our experiments seems lower. The experimental analy- |

| | | |
|---|---|---|
| | espect values in the order of 1700 - 1900 kg/mˆ3. | sis here is considered to be the preliminary achievements. The authors appreciate the reviewer's valuable suggestions to carry out more experiments involving debris-flow densities in the order of 1700 - 1900 kg/mˆ3 in the future. |
| **7.** | #1: indicate flow direction and exchange the word "behind" with "downriver of". | The word "behind" has been replaced by "downriver of". |
| **8.** | #2a: Sabo dam is never use in the text. Use check dam or replace check dam with sabo dam in the text. | The word "Sabo dam" has been replaced by "check dam". |
| **9.** | #5: desribe it as "debris-flow hydrograph". If your LRF gave you min, max and mean values, you could perhaps explain the outliers. And: this hydrograph does not really show a typicall steep front of a debris flow. It looks more like a hyperconcentraded flood. Again: add information on the sampling rate of the device. | The caption of Figure 5 was changed to "debris-flow hydrograph". The information on the sampling rate of the device was added in line 90, page 5. |
| **10.** | #6: add an arrow to show flow directions. Very small images. Perhaps increase contrast. | An arrow in each figure was added to show debris-flow directions in Fig.6. |
| **11.** | Scaling effects are not discussed. Please add a section to explain how the results of the experiments can be use in real dimensions. What is the Froude number of your experiments? | The scaling effects are discussed in the revised vision in lines 207-215, page 11. The Froude number in our experiment ranged from 1.14 to 1.16. It meant that the debris flows in the experiments were supercritical flow (in lines91-92, page 5). |
| **12.** | I miss a sensitivity study on different debris-flow mixtures (e.g. higher densities, water content variations) | The variation of debris-flow density (different debris-flow mixtures) on scour depth was added in lines 135-141, page 7. |
| **13.** | I miss information on the LRF. What is the sampling rate (in Hz) of the device? How are | The information on the LRF was given in lines 89-90, page 5 and the sampling rate |

| | |
|---|---|
| splashing effects handled? | (Frequency) was added in line 90, page 5. |
| **14.** What would happen, if there is driftwood involved? Did you test that or what do you expect in such a case? | Debris flows with driftwood will speed up the blockage and jamming of a check dam. Provided that driftwood is involved in our experiments, the check dam will capture driftwood when it passed through the spillway with debris flows. The subsequent debris flows will overflow from the check dam crest once the spillway is blocked by the driftwood, which will cause scour downriver of a check dam. The debris flows with driftwood was not considered in the current experiments, but definitely the reviewer has raised a very important question. The related experiments will be carried out to investigate the behaviour of debris flows with driftwood and its scour feature in the future. |
| **15.** Can you say something about abrasion rates and the expected life time of such structures? | Abrasion occurs due to the interaction between solid particles in debris flows and the boundary of hydraulic structures. For a spillway with curved bottom, the reaction of centrifugal force exerting on spillway bottom enhance the interaction between the solid particles and the bottom (a component of the reaction force has the same direction as the gravitational force of debris flows near the outlet of the spillway). Although abrasion phenomenon is common, it is difficult to quantify the abrasion rate during an episode of debris flows.
Abrasion may be one of the factors lead to the |

| | damage of spillway with lateral contraction. However, some methods can be taken to mitigate the abrasion damage of such structures by using anti-abrasion materials, or add the protecting layer. The check dam with lateral contracted spillway, like other check dams, the expected life time mainly depends on the debris-flow scales, flow velocity, particle concentration, etc. |
| --- | --- |

[revised manuscript text omitted]

**4.2 Discussions**

The characteristics of debris flow nape and scour downriver of a check dam with different spillway were experimentally investigated in this article. When the experimental data are used to predict debris-flow motion and scour feature downriver of a check dam in prototype, the effects of physical model scale should be considered. Scaling effect is mainly induced by dissatisfaction of mobility similitude of model sediment in physical model experiments and it leads to discrepancies between the estimated and actual scour results. Just like the experimental investigation on the scale effect in pier-scour experiments, the bed-particle mobility similitude (Ettema et al ,1998; Ettema and Melville,1999) or the flow-strength similitude (Lee and Sturm,2009) should be satisfied to weaken or eliminate the scaling effect for debris-flow scour when the experimental results are extrapolated to predict prototype performance in the future.

When debris flows occur in the mountainous areas with forest the driftwood carried by debris flows is a common phenomenon. The debris flows combined with driftwood will speed up blockage and jamming of a check dam. Once the spillway is blocked by the driftwood the subsequent debris flows will overflow from the crest of a check dam, which will cause extensive scour downriver of a check dam. Therefore, it is also necessary to investigate the behavior of debris flows with driftwood and propose some reasonable structural or non-structural countermeasures to mitigate the effects of debris flows with driftwood on the operation of a check dam in the future.

**Acknowledgments**

The study results presented in this paper were supported by the Key Research Program of the Chinese Academy of Sciences (Grant No. KZZD-EW-05-01), the National Natural Science Foundation of China (Grant No.51209195), the Science Technology Service Network Initiative, Chinese Academy of Sciences (Grant No. KFJ-EW-STS-094), and the Key Laboratory of Mountain Hazards and Earth Surface Process, Chinese Academy of Sciences.

**List of symbols**

ai $=$ The index for the dimensionless parameter (-)

$b$ $=$ The width of the spillway outlet(m)

$B$ $=$ The width of the spillway inlet (m)

$d_{90}$ $=$ The characteristic particle size for erodible bed materials (m)

$k$ $=$ The coefficient for the dimensionless equation (-)

$h_1$ $=$ The depth of debris flows at reference cross-sections #1 (m)

$h_2$ $=$ The depth of debris flows at reference cross-sections #2 (m)

$h_d$ $=$ The maximum scour depth (m)

$h_w$ $=$ The water head loss (m)

$g$ $=$ The acceleration of gravity (m/s$^2$)

$q$ $=$ The unit discharge of the debris flow (m$^3$/s)

$v$ $=$ The initial velocity of the debris flow flowing out of the spillway(m/s)

$v_1$ $=$ The velocity of debris flows at reference cross-sections #1 (m/s)

$v_2$ $=$ The velocity of debris flows at reference cross-sections #2 (m/s)

$V$ $=$ The scale of debris flow in the experiments (m$^3$)

$x$ $=$ Trajectory in the horizontal direction (m)

$y$ $=$ The water elevation difference (m)

$Z_1$ $=$ The elevation of reference cross-sections at #1 (m)

$Z_2$ $=$ The elevation of reference cross-sections at #2 (m)

$\Delta z$ $=$ The elevation difference between the two reference cross-sections (m)

Greek letters

$\alpha_1$ $=$ The kinetic energy correction coefficient for $v_1$ (-)

$\alpha_2$ $=$ The kinetic energy correction coefficient for $v_2$ (-)

$\rho_d$ $=$ The density of debris flows (kg/ m$^3$)

$\rho_w$ $=$ The density of clear water (kg/ m$^3$)

$\zeta$ $=$ The energy dissipation coefficient(-)

$\eta$ $=$ The lateral contraction ratio(-)

$\varphi$ $=$ The angle of the initial velocity in the horizontal direction($^\circ$)

[revised manuscript text omitted]

Although a high lateral contraction ratio extended the debris-flow nappe, it also increased the scour depth in the erodible bed to some extent. In addition, debris-flow density has some effects on the scour depth. Figure 11

indicates the scour depth caused by debris flow with density of 1200kg/m$^3$ is a bit larger than that caused by debris flow with density of 1500kg/ m$^3$ at a certain lateral contraction ratio (Figure 11). It was explained that the debris flow with lower particle concentration (Lower debris-flow density) initialized and carried more bed materials than that with higher particle concentration (Higher debris-flow density) when the other factors were fixed (Such as longitudinal slope of gully, debris-flow scale, lateral contraction ratio of the spillway).

[revised manuscript text omitted]

**4.2    Discussions**

The characteristics of debris flow nape and scour downriver of a check dam with different spillway were experimentally investigated in this article. When the experimental data are used to predict debris-flow motion and scour feature downriver of a check dam in prototype, the effects of physical model scale should be considered. Scaling effect is mainly induced by dissatisfaction of mobility similitude of model sediment in physical model experiments and it leads to discrepancies between the estimated and actual scour results. Just like the experimental investigation on the scale effect in pier-scour experiments, the bed-particle mobility similitude (Ettema et al ,1998; Ettema and Melville,1999) or the flow-strength similitude (Lee and Sturm,2009) should be satisfied to weaken or eliminate the scaling effect for debris-flow scour when the experimental results are extrapolated to predict prototype performance in the future.

When debris flows occur in the mountainous areas with forest the driftwood carried by debris flows is a common phenomenon. The debris flows combined with driftwood will speed up blockage and jamming of a check dam. Once the spillway is blocked by the driftwood the subsequent debris flows will overflow from the crest of a check dam, which will cause extensive scour downriver of a check dam. Therefore, it is also necessary to investigate the behavior of debris flows with driftwood and propose some reasonable structural or non- structural countermeasures to mitigate the effects of debris flows with driftwood on the operation of a check dam in the future.

**Acknowledgments**

The study results presented in this paper were supported by the Key Research Program of the Chinese

Academy of Sciences (Grant No. KZZD-EW-05-01), the National Natural Science Foundation of China (Grant

No.51209195), the Science Technology Service Network Initiative, Chinese Academy of Sciences (Grant No.

KFJ-EW-STS-094), and the Key Laboratory of Mountain Hazards and Earth Surface Process, Chinese Academy of Sciences.

**List of symbols**

$\text{ai}$ = The index for the dimensionless parameter (-)
$b$ = The width of the spillway outlet(m)
$B$ = The width of the spillway inlet (m)
$d_{90}$ = The characteristic particle size for erodible bed materials (m)
$k$ = The coefficient for the dimensionless equation (-)
$h_1$ = The depth of debris flows at reference cross-sections #1 (m)
$h_2$ = The depth of debris flows at reference cross-sections #2 (m)
$h_d$ = The maximum scour depth (m)

$h_w$ = The water head loss (m)
$g$ = The acceleration of gravity (m/s$^2$)
$q$ = The unit discharge of the debris flow (m$^3$/s)
$v$ = The initial velocity of the debris flow flowing out of the spillway(m/s)
$v_1$ = The velocity of debris flows at reference cross-sections #1 (m/s)
$v_2$ = The velocity of debris flows at reference cross-sections #2 (m/s)
$V$ = The scale of debris flow in the experiments (m$^3$)
$x$ = Trajectory in the horizontal direction (m)
$y$ = The water elevation difference (m)
$Z_1$ = The elevation of reference cross-sections at #1 (m)
$Z_2$ = The elevation of reference cross-sections at #2 (m)
$\Delta z$ = The elevation difference between the two reference cross-sections (m)

Greek letters

$\alpha_1$ = The kinetic energy correction coefficient for $v_1$ (-)
$\alpha_2$ = The kinetic energy correction coefficient for $v_2$ (-)
$\rho_d$ = The density of debris flows (kg/ m$^3$)
$\rho_w$ = The density of clear water (kg/ m$^3$)
$\zeta$ = The energy dissipation coefficient(-)
$\eta$ = The lateral contraction ratio(-)
$\varphi$ = The angle of the initial velocity in the horizontal direction($^\circ$)

[revised manuscript text omitted]

---

## Referee Comment (RC2) · Anonymous Referee #2 · 27 Sep 2016

This manuscript presents experiments and an empirical model on debris flow and scour features behand a check dam. The paper is well written and the scope and contents of the manuscript are appropriate for the journal. Therefore, I can recommend this manuscript to be accepted after minor revisions.

The following questions need to be clarifiedïijŽ

In the experiments, the size of the model is much smaller than in the reality, which leads to much smaller stress in the debris flow and check dam. How would the results change for large scaled models? Please add some discussions.

Page 8 line 151 The mean value of the energy dissipation rate demonstrated a good

positive correlation between the energy dissipation rate and the lateral contraction ratio. Do you mean: The mean value of the energy dissipation rate demonstrated a good positive correlation with the lateral contraction ratio?

Page 9 line 178 The absolute error was smaller than 15.0% in most cases, as shown in Figure 11 . What means in most cases, how many percent?

---

## Author Comment (AC2) · 7 Oct 2016

1. In the experiments, the size of the model is much smaller than in the reality, which leads to much smaller stress in the debris flow and check dam. How would the results change for large scaled models? Please add some discus-sions. Answer:thanks very much for the reviewer's com-ment. Some discussions were added in the revised manuscript (in lines 203-205, page 10;212-213, page 11). 2. Page 8 line 151 The mean value of the energy dissipation rate demonstrated a good, positive correlation between the energy dissipation rate and the lateral contraction ratio. Do you mean: The mean value of the energy dissipation rate demonstrated a good positive correlation with the lateral contraction ratio? Answer:the wrong sentence has been revised

as" The mean value of the energy dissipation rate demonstrated a good positive correlation with the lateral contraction ratio"(in line 157, Page 8). 3. Page 9 line 178 The absolute error was smaller than 15.0% in most cases, as shown in Figure 11 . What means in most cases, how many percent? Answer:the exact value was given in the revised manuscript (in line183, page 9) .

Please also note the supplement to this comment:
http://www.nat-hazards-earth-syst-sci-discuss.net/nhess-2016-189/nhess-2016-189-AC2-supplement.pdf

**Supplement:**

We feel very grateful to the reviewer who has given us the valuable suggestions and comments for our paper. We have revised our manuscript accordingly.

Huayong Chen

Responses to the reviewer' comments:

| Comments of Anonymous Referee #2: | Author's Reply |
|---|---|
| **1.** In the experiments, the size of the model is much smaller than in the reality, which leads to much smaller stress in the debris flow and check dam. How would the results change for large scaled models? Please add some discussions. | Thanks very much for the reviewer's comment. Some discussions were added in the revised manuscript (in lines 203-205, page 10;212-213, page 11). |
| **2.** Page 8 line 151 The mean value of the energy dissipation rate demonstrated a good, positive correlation between the energy dissipation rate and the lateral contraction ratio. Do you mean: The mean value of the energy dissipation rate demonstrated a good positive correlation with the lateral contraction ratio? | The wrong sentence has been revised as" The mean value of the energy dissipation rate demonstrated a good positive correlation with the lateral contraction ratio"(in line 157, Page 8). |
| **3.** Page 9 line 178 The absolute error was smaller than 15.0% in most cases, as shown in Figure 11 . What means in most cases, how many percent? | The exact value was given in the revised manuscript (in line183, page 9) . |

---

## Author Comment (AC3) · 20 Oct 2016

Question 1: In the experiments, the size of the model is much smaller than in the reality, which leads to much smaller stress in the debris flow and check dam. How would the results change for large scaled models? Please add some discussions. Answer: Thanks very much for the reviewer's comment. The size of a model indeed is an important parameter for experimental design. When we choose a bigger experimental model, the hydraulic phenomena or results obtained in the experiment will be closer to the data in prototype. Generally, rational model scale will be chosen to simulate the movement of debris flows or bed erosion. In our experiments, we simulated the debris flow patterns and scour features downriver of a check dam at a certain scale.

[Figure]

As the reviewer says, the stress between the debris flows and check dam may be smaller than that in reality. However, the experimental results can still account for the interaction between the debris flows and erodible bed under certain hydraulic structure. Based on the similitude principle, the experimental results obtained in the small scaled model can be extrapolated for large scaled models. Some discussions were added in the revised manuscript (in lines 203-205, page 10;212-213, page 11).

Question 2: Page 8 line 151 The mean value of the energy dissipation rate demonstrated a good, positive correlation between the energy dissipation rate and the lateral contraction ratio. Do you mean: The mean value of the energy dissipation rate demonstrated a good positive correlation with the lateral contraction ratio. Answer: This sentence was not good enough. It has been replaced by" The mean value of the energy dissipation rate demonstrated a good positive correlation with the lateral contraction ratio" (in line 157, Page 8).

Question 3: Page 9 line 178 The absolute error was smaller than 15.0% in most cases, as shown in Figure 11 . What means in most cases, how many percent? Answer: The phase "In most cases" in the manuscript is not accurate enough. It has been replaced by the exact value (8.33%) calculated based on the calibration data (in line183, page 9).

Please also note the supplement to this comment:
http://www.nat-hazards-earth-syst-sci-discuss.net/nhess-2016-189/nhess-2016-189-AC3-supplement.pdf
* * *